# Modulation of Reoviral Cytolysis (I): Combination Therapeutics

**DOI:** 10.3390/v15071472

**Published:** 2023-06-29

**Authors:** Yoshinori Mori, Sandra G. Nishikawa, Andreea R. Fratiloiu, Mio Tsutsui, Hiromi Kataoka, Takashi Joh, Randal N. Johnston

**Affiliations:** 1Cumming School of Medicine, University of Calgary, Calgary, AB T2N 1N4, Canada; ysnmori@yahoo.co.jp (Y.M.); snishika@ucalgary.ca (S.G.N.); arfratil@ucalgary.ca (A.R.F.);; 2Graduate School of Medical Sciences, Nagoya City University, Nagoya 467-8601, Japan

**Keywords:** reovirus, oncolytic virus, oncolytic viral therapy, Ras pathway, chemotherapy

## Abstract

Patients with stage IV gastric cancer suffer from dismal outcomes, a challenge especially in many Asian populations and for which new therapeutic options are needed. To explore this issue, we used oncolytic reovirus in combination with currently used chemotherapeutic drugs (irinotecan, paclitaxel, and docetaxel) for the treatment of gastric and other gastrointestinal cancer cells in vitro and in a mouse model. Cell viability in vitro was quantified by WST-1 assays in human cancer cell lines treated with reovirus and/or chemotherapeutic agents. The expression of reovirus protein and caspase activity was determined by flow cytometry. For in vivo studies, athymic mice received intratumoral injections of reovirus in combination with irinotecan or paclitaxel, after which tumor size was monitored. In contrast to expectations, we found that reoviral oncolysis was only poorly correlated with Ras pathway activation. Even so, the combination of reovirus with chemotherapeutic agents showed synergistic cytopathic effects in vitro, plus enhanced reovirus replication and apoptosis. In vivo experiments showed that reovirus alone can reduce tumor size and that the combination of reovirus with chemotherapeutic agents enhances this effect. Thus, we find that oncolytic reovirus therapy is effective against gastric cancer. Moreover, the combination of reovirus and chemotherapeutic agents synergistically enhanced cytotoxicity in human gastric cancer cell lines in vitro and in vivo. Our data support the use of reovirus in combination with chemotherapy in further clinical trials, and highlight the need for better biomarkers for reoviral oncolytic responsiveness.

## 1. Introduction

Nearly one million cases of gastric cancer are diagnosed worldwide each year, with the highest incidence occurring in eastern Asia, parts of South America and eastern Europe. Gastric cancer is the second most common cause of cancer-related death worldwide, with around 700,000 deaths a year. In the US, more than 20,000 new cases of gastric cancer and 10,000 deaths occur annually [1]. Survival of patients with gastric cancer is substantially worse than that of patients with most other solid malignancies. The only current treatment that offers potential cure is complete resection of the tumor [2]. However, after complete resection alone, patients who are shown to have extensive lymph node involvement on surgical pathology specimens have a 5-year survival of only 7% [3]. Recently, the use of chemotherapy such as cisplatin, irinotecan, paclitaxel, or docetaxel for gastric cancer has yielded some improvements in outcome, but the beneficial effects are still modest and there is a real need for new therapeutic strategies [4]. It is possible that oncolytic virus therapy may be useful in this context, and perhaps especially so for viruses that proliferate in the gastrointestinal tract.

Human reovirus is a ubiquitous, non-enveloped virus containing 10 segments of double-stranded RNA (dsRNA) as its genome, with infections that are generally mild, restricted to the upper respiratory and gastrointestinal tracts, and often asymptomatic [5]. Reovirus has innate oncolytic potential in a wide range of murine and human tumor cells, and this is at least partly dependent on the transformed state of the cell [6,7]. The precise mechanism of reoviral tropism and selective oncolysis in malignant cells is yet to be fully determined. In normal cells, the presence of an intact double-stranded RNA-activated protein kinase system limits the establishment of a productive reoviral infection. In malignant cells with an activated Ras pathway, it has been suggested that either directly through Ras mutation or indirectly via upregulation of epidermal growth factor receptor (EGFR) signaling or of other signaling components [5,8,9,10,11], this cellular antiviral response mechanism may be perturbed, viral replication enhanced, and subsequent lysis of the host cancer cell facilitated. The modulation of other pathways that regulate viral attachment, penetration, uncoating, assembly, and propagation further influence the efficiency of viral oncolysis (see below).

Thus, reovirus is considered a promising candidate for oncolytic virus therapy and Phase II and III clinical studies are underway (oncolyticsbiotech.com (accessed on 5 June 2023)) for multiple tumor categories, and an orphan drug designation has recently been approved by the FDA for the use of reovirus in the treatment of gastric and certain other cancers [12]. While monotherapy with oncolytic reovirus has been explored, most current efforts use reovirus or other oncolytic viruses such as herpes virus in combination with chemotherapy or radiotherapy in preclinical or clinical studies to potentially increase treatment efficacy for various malignancies [13,14,15,16,17,18]. For example, several studies have suggested that reovirus, in combination with paclitaxel against lung cancer and with cisplatin against melanoma, may have synergistic antitumoral effects [19,20]. However, so far there have been few reports of oncolytic reovirus and combination therapy against gastric cancer in preclinical animal models [21]. Here we show that oncolytic reovirus therapy is effective against gastric cancer xenografts in a mouse model, and that the combination of reovirus and chemotherapeutic agents paclitaxel and irinotecan has clear synergistic benefits.

## 2. Materials and Methods

### 2.1. Cells and Virus

Human gastric cancer cell lines (KATOIII, SNU16, AGS, NCI-N87, Hs746T) and human colon cancer cell lines (HCT116, HT-29) were obtained from the American Type Culture Collection (ATCC; Rockville, MD, USA). Other human gastric cancer cell lines (MKN1, MKN7, MKN45, MKN74, HGC-27, GCIY) were obtained from the Cell Bank, RIKEN BioResource Center (Kyoto, Japan). Another human gastric cancer cell line (FU97) was obtained from the Health Science Research Resources Bank (Japan), and ISt-1 was a gift from Dr. Masanori Terashima. All cell lines were tested and free of mycoplasma contamination. The Dearing strain of reovirus serotype 3 (a gift from P. Lee, Dalhousie University, Halifax, NS, Canada) was propagated in suspension cultures of L929 cells (from ATCC) and purified according to previously established methods [8,9] with the exception that β-mercaptoethanol was omitted from the extraction buffer. Viral titers were also established using L929 cells [9].

### 2.2. Chemotherapeutic Agents

Irinotecan (Mayne Pharma, Raleigh, NC, USA), Paclitaxel (Hospita, Canada), and Docetaxel (Taxotere^®^; Sanofi Aventis, Bridgewater, NJ, USA) were kindly provided by Dr. Aru Narendran, University of Calgary. These agents were diluted with the respective medium just before use for in vitro studies and with phosphate-buffered saline (PBS) for in vivo studies.

### 2.3. Cell Viability Assay

All cells were seeded in 96-well plates at a density of 2 × 10^3^ cells/well with appropriate medium. GC cells were mock infected or infected with reovirus at a MOI of 1 or 10 and then treated with chemotherapeutic agents. Experiments were repeated three times and results presented as mean +/− standard deviation. Numbers of viable cells were evaluated by a colorimetric WST-1 assay at 3, 6, and 9 days post treatment. WST-1 (Roche), a tetrazolium salt, is cleaved to a colored formazan product by enzymes in metabolically active cells, and the reaction is quantitated with an automatic plate reader at 450 nm. The potential synergistic effect arising from the combination of reovirus with chemotherapy on cell proliferation was assessed by calculating combination index (CI) values using the method of Zhao et al. [22]. The CI provides a quantitative measure of the degree of interaction between two or more agents. A CI of 1 denotes an additive interaction, >1 represents antagonism, and <1 indicates synergy, with lower values indicating a higher degree of augmentation of effect of the two agents working together.

### 2.4. Ras Activation Assay

First, 85–90% confluent cells grown in 150 mm dishes were lysed with 1 × Mg^2+^ lysis buffer (Ras activation assay kit; Millipore). To determine the level of activated Ras (Ras-GTP) in these cells, 1 mg of cell lysate was incubated with 10μL of Raf-1 Ras binding domain agarose conjugate at 4 °C for 45 min. The beads were then collected, washed, resuspended in 2× Laemmli buffer, and boiled for 5 min. This was then followed by SDS-PAGE and Western blotting with an anti-Ras antibody (clone RAS 10) according to the manufacturer’s instructions. To determine the level of total Ras, cell lysates were directly subjected to SDS-PAGE and Western blotting with anti-Ras antibody. The membrane was incubated with horseradish peroxidase-conjugated goat antimouse antibody, and specific bands were detected with an ECL system (GE Healthcare).

### 2.5. FACS Analysis

After treatment with reovirus and/or chemotherapeutic agents, general caspase activity was assessed by the carboxyfluorescein caspase detection kit (Apologix; cat. no. FAM100-2; Cell Technology, Inc., Newport News, VI, USA), which is based on carboxyfluorescein-labeled fluoromethyl ketone (FMK)—peptide inhibitors of caspases. These inhibitors are cell permeable and noncytotoxic. Once inside the cell, the fluorescent inhibitor binds covalently to the active caspase. Primary rabbit antireovirus polyclonal antibody was made in our lab and detected by binding to PE goat anti-rabbit IgG (Cedarlane Laboratories Ltd., Burlington, ON, Canada). Fixed and permeabilized cells were analyzed by flow cytometry.

### 2.6. Subcutaneous Tumor Xenograft Model in Nude Mice

Six-week-old male CD-1 nude mice, purchased from Charles River, were kept under pathogen-free conditions according to a protocol approved by the University of Calgary Animal Care Committee. MKN45 cells (2 × 10^6^) were implanted subcutaneously in the left flanks of mice under anesthesia. When the tumors reached a diameter of ~5 mm, the mice were randomly divided into four groups (5 mice/group), and a 50 μL solution containing reovirus (1 × 10^8^ PFU/animal) or PBS was injected into the tumor (any excess injected fluid was distributed in surrounding tissues). Simultaneously, each mouse received an intraperitoneal injection of 100 μL paclitaxel at a dose of 10 mg/kg or irinotecan at a dose of 5 mg/kg. The tumor size was calculated by external caliper measurements every 2 or 3 days. Tumor volume was calculated using the following formula: tumor volume (mm^3^) = *a* × *b*^2^ × 0.5, where *a* is the longest diameter, *b* is the shortest diameter, and 0.5 is a constant to calculate the volume of an ellipsoid. Statistical differences among groups were assessed using the Mann–Whitney *U* test.

### 2.7. Immunodetection of Reoviral Replication

For histological analysis, tumors were fixed in 10% neutral buffered formalin, embedded in paraffin, and sectioned. Sections were then immersed in xylene, followed by rehydration in decreasing concentrations of ethanol. Endogenous peroxidase was inactivated in 3% hydrogen peroxide in PBS for 15 min. Sections were then incubated in primary rabbit antireovirus polyclonal antibody (1:1000 in PBS with 10% goat serum and 0.3% Triton X-100) partially purified by ammonium sulfate precipitation. Slides were washed in PBS and then subjected to avidin-biotin horseradish peroxidase staining as recommended by the manufacturer (Vector, Burlington, ON, Canada) and counterstained in hematoxylin.

## 3. Results

### 3.1. Reovirus Cytotoxicity in Gastric Cancer Cell Lines

We first surveyed the cytotoxicity of reovirus against gastric cancer using the WST-1 assay in 14 different human gastric cancer cell lines. After 72 h exposure, reovirus alone showed moderate cytopathic effects (relative cell viability was between 0.2 and 0.8) in six gastric cancer cell lines (AGS, MKN1, NCI-N87, Hs746T, FU97, ISt-1) and low cytopathic effects (relative cell viability was more than 0.8) in seven gastric cancer cell lines (HGC-27, KATOIII, MKN7, MKN45, MKN74, NUGC4, SNU16). Only GCIY cells showed a high cytopathic effect (relative viability was less than 0.2; Figure 1a).

To determine whether the variable cellular responses to reovirus might be explained by differing levels of Ras activation, we then measured the levels of GTP-Ras in the various gastric cancer cell lines. Activated Ras was detectable in most gastric cancer cell lines (Figure 1b) when compared with a negative control of normal human fibroblasts. However, we observed no obvious correlation between Ras activity levels and cytolytic effects—some gastric cancer cell lines with prominent Ras activation were relatively resistant to reovirus (e.g., AGS), whereas the most responsive cell line (GCIY) displayed only very modest activation of GTP-Ras. While variable cytolytic responses to reovirus may reflect multiple cellular features (abundance of receptor, efficiency of viral uncoating, etc.), it is clear that in these gastric cancer cell lines, variables beyond simple Ras pathway activation must be modulating cellular responses to reovirus infection.

### 3.2. Reovirus Cytotoxicity with Chemotherapeutic Agents in Gastric Cancer Cell Lines

We then chose four different gastric cancer cell lines (GCIY, AGS, NCI-N87, and MKN45, showing strong, medium, or minimal responses to reovirus) to examine cytotoxicity in more detail. We evaluated cell viability with combinations of reovirus and chemotherapeutic agents using WST-1 assays at days 3, 6, and 9 after treatment. We chose irinotecan, paclitaxel, and docetaxel as combination chemotherapeutic agents, as these are already commonly used in treatments of human patients. Although the various cell lines showed modest differences in responses to the chemotherapeutic drugs alone, the various agents showed clear enhancement of cell killing when supplemented with reovirus in MKN45 and AGS cells (Figure 2a,b).

Reovirus alone killed GCIY very well, so we could not evaluate synergy with chemotherapy in these cells, and NCI-N87 cells showed no enhancement in combination experiments (Appendix A). All Combination Indices for MKN45 and AGS cells were less than 1, which therefore showed synergy of reovirus with chemotherapy (Figure 2d); enhanced killing was also revealed in photomicrographs of these two cell populations (Figure 2c).

### 3.3. Combinations of Reovirus and Chemotherapeutic Agents Enhanced Reovirus Replication and Apoptosis

We then tested whether the administration of chemotherapeutic agents might enhance or diminish viral activity, while promoting cell death. For this purpose, we evaluated reoviral protein expression and caspase activity using FACS analysis of MKN45 and AGS cells treated in eight groups (control, reovirus alone, irinotecan alone, paclitaxel alone, docetaxel alone, reovirus and irinotecan, reovirus and paclitaxel, and reovirus and docetaxel). Nearly every combination of reovirus and chemotherapeutic agents enhanced reoviral protein synthesis in these cells, which as we have shown previously leads to the elevated release of infectious viral particles [11]. Reovirus or chemotherapy alone enhanced caspase activity to some extent, and this was enhanced further when reovirus and chemotherapy were combined (Figure 3).

In GCIY and NCI-N87 cells, reovirus alone enhanced caspase activity in both cell types (Appendix A). These results show that the combination of reovirus and chemotherapeutic agents can enhance reovirus protein synthesis in some cell lines and induces effects leading to cell death via apoptosis or possibly pyroptosis.

### 3.4. Combined Reovirus and Chemotherapeutic Agents Enhanced Anti-Tumor Effects in a Murine Gastric Cancer Xenograft Model

We then assessed the therapeutic efficacy of reovirus in combination with chemotherapeutic agents against gastric cancer cells in vivo. We made two types of gastric cancer xenograft models, with CD-1 nude mice bearing either MKN45- or GCIY-based tumors. Then, we treated the former MKN45 model with combination therapy in four groups (control, reovirus alone, chemotherapeutic agent (irinotecan or paclitaxel) alone, and reovirus plus chemotherapeutic agent); the latter model was treated with monotherapy in two groups (control and reovirus alone). Administration of reovirus, irinotecan, or paclitaxel results in significant tumor growth suppression compared with the untreated control at 28 days after initiation of treatment. Importantly, the combination of reovirus plus irinotecan or reovirus plus paclitaxel produced a more profound inhibition of tumor growth compared with mice treated with either modality alone and control (Figure 4A–D).

In the GCIY model, reovirus monotherapy was effective and sufficient when compared with the untreated control (Appendix A), similar to the effect observed in vitro. Extensive viral distribution in the MKN45 tumors was confirmed by immunohistochemical staining of reovirus protein (Figure 4E). There were no significant differences in the mean body weights among experimental groups, and no morbidity was attributable to therapy with reovirus, paclitaxel, irinotecan, or both in combination.

## 4. Discussion

Many oncolytic viruses have been developed for application in multiple kinds of cancer. However, there are relatively few reports about the potential use of oncolytic viruses with gastric cancer. For example, adenoviral vectors have been used for experimental treatment and gene therapy of various cancers because of their high transduction efficiency. However, adenoviral infectivity of gastrointestinal cancer cells is generally poor due to the limited expression of the coxsackie-adenovirus receptor [23]. Thus, we were interested in exploring the use of reovirus in gastric cancer, as it naturally replicates in the gastrointestinal tract and its cell surface receptor (JAM-A) is abundant.

One of our first questions was therefore to determine the level of activation of Ras signaling in our gastric cancer cells, as this has been reported in other systems to correlate with susceptibility to reoviral oncolysis [5,8]. Although oncogenic mutations of Ras are infrequent (2–7%) in gastric cancer [24]; we nevertheless found evidence for variably activated GTP-Ras in most of the gastric cancer cell lines we used in this experiment (Figure 1b), possibly due to upstream activation of receptors such as EGFR, which is often mutated in gastric cancer. Thus, it is plausible for this reason that reovirus can be used as a gastric cancer therapy, even though we were surprised by the poor correlation between the levels of activated GTP-Ras and cytolytic susceptibility to reovirus (compare Figure 1a,b). The weakness of this association has also been reported by others [25] and it is likely that in addition to the activation status of Ras-associated pathways [5,26], there are other molecular determinants of reovirus-sensitivity, such as the cellular abundance of putative reovirus receptors/coreceptors [27,28,29], intracellular virion uncoating or assembly processes [30,31], and viral release and propagation, all of which can affect reovirus infection and oncolytic efficiency. In addition, as we propose in our accompanying manuscript, it is possible that the degree of cellular stemness may vary among different cancers, and this may confer variable reoviral responsiveness.

The cytopathic effect of reovirus we observed with our gastric cancer cell lines was relatively modest when compared with other gastrointestinal cancers (colon, esophageal, liver, pancreas; in preparation). GCIY cells were very susceptible to reovirus, but most gastric cancer cell lines showed moderate or low cytopathic effects in vitro (Figure 1a). Thus, from this evidence, it would be difficult to justify the use of reovirus as monotherapy against gastric cancer. However, we wished to consider whether some combination of reovirus and chemotherapy might show synergistic effects in gastric cancer and perhaps expand the range of gastric cancers in which benefits could be achieved.

Many preclinical studies have provided experimental evidence for effective killing of cancer cells by oncolytic viruses [32,33,34,35,36]. In animal models, however, established xenograft tumors are rarely eliminated despite the existence of persistently high viral titers within the tumor, and it is possible that total elimination of solid tumors may require higher doses of oncolytic viruses that might prove toxic or lethal. In a report of a clinical trial of ONYX-015 adenovirus, no clinical benefit was noted in the majority of patients, despite encouraging biological activity [37]. Tumor progression was rapid in most patients, even though substantial necrosis was noted in the tumors after treatment [38,39]. Thus, we opted to evaluate chemovirotherapy, consisting of oncolytic virotherapy combined with low doses of a chemotherapeutic agent. We reasoned that sublethal doses of chemotherapy might damage cancer cell pathways and reduce, for example, innate anti-viral responses, thereby enhancing viral oncolysis while reducing the likelihood of adverse effects [40]. In this study, we chose to explore the co-administration of irinoctecan, paclitaxel, and docetaxel because these are often used for second-line chemotherapy in human patients and are therefore plausible choices for future clinical trials employing chemovirotherapy.

Indeed, several chemotherapy/oncolytic virus combinations have already been evaluated to date and have been shown to result in enhanced antitumor effects without compromising safety. For example, the adenovirus Onyx-015 enhanced clinical efficacy by combining intratumoral Onyx-015 with systemic cisplatin and 5-fluorouracil when compared with chemotherapy alone [41]. E1A-expressing adenoviral E3B mutants combined with cisplatin and paclitaxel [42] also showed synergistic activity in vitro and in vivo. Combinations using oncolytic herpesviruses, such as G207 with cisplatin and HSV-1716 with mitomycin C, resulted in synergistic activity in vitro and in vivo [43,44,45]. Even though the precise biochemical mechanisms by which these synergies are achieved remain unknown, their potential use in gastric cancer may provide new options for more successful treatment.

In the experiments described here, we found that irinotecan, paclitaxel, and docetaxel were all able to enhance reovirus replicative activity (Figure 3) in MKN45 and AGS cells. It is possible that these chemotherapeutic agents act by repressing the innate immune responses of cells, thereby enhancing virus replication. In any case, it appears that the elevated viral replication is linked to greater caspase activation and thus an acceleration in programmed cell death. 

In our experiments with xenograft tumors in vivo, we did not expect reovirus alone to be very effective in repressing growth in the MKN45 gastric cancer model, simply because reovirus did not kill MKN45 cells very well in vitro. However, we found that even reovirus by itself was able to strongly repress (though not completely eliminate) growth of these tumor cells in vivo. Nevertheless, both combinations of reovirus plus paclitaxel or reovirus plus irinotecan, showed clear enhancement of cell killing in vivo (Figure 4), consistent with our results in vitro.

Thus, we conclude that our in vitro and in vivo data both encourage further studies of reovirus plus chemotherapy (and especially with irinotecan and paclitaxel, which were superior to docetaxel) as a viable therapeutic modality for gastric cancer, even though in some cases (Appendix A and reference [46]), reovirus monotherapy may be partly or fully effective. Our results are obtained in immunodeficient mice, and thus studies with immunocompetent hosts and syngeneic tumors [47] could show further enhancement in tumor cell killing. In other work [48], we found that trastuzumab is able to enhance reoviral oncolysis in gastric cancers that overexpress the Her2/neu oncogene. Thus, we argue that there is a clear cellular rationale for combining chemotherapeutic agents and oncolytic reovirus for the treatment of this disease, even though the precise mechanisms underlying synergy between reoviral oncolysis and specific chemotherapeutic drugs remain unclear. We also note the potential for different routes of viral administration, such as intraperitoneal or intravascular, or especially for tumors of the gastrointestinal tract, via direct administration orally or as we have shown anally [49]. As more is learned about the molecular pathways by which oncolytic reovirus kills many cancer cell types, while sparing normal cells and tissues, the rational combination of more potent agents or immunomodulators ([50,51,52,53,54]; also Kubota et al., submitted) plus the potential for engineered virus [55] with greater anticancer action and reduced side effects will become clearer. Finally, we return to the observation that we (Figure 1) and others have made, which is that the correlation between Ras activation and reoviral responsiveness is unexpectedly poor. Transfection of activated Ras genes into normal cells can indeed result in both cellular transformation and reoviral susceptibility, as originally reported by Lee’s group [26], but many tumor cells with activated Ras may display reoviral resistance, whereas in other cases, those with low Ras pathway activity may still be sensitive to the virus. Ras pathway activation alone is therefore unlikely to be a reliable biomarker for reoviral responsiveness, which as we have argued is indeed subject to multiple cellular constraints [56]. In our accompanying manuscript, we propose that a previously unappreciated variable, that of cellular stemness in cancer (or embryonic) cells, may be a novel important factor in predicting reoviral responsiveness. Further work will be required to evaluate more fully the utility of this proposed relationship.

In our accompanying manuscript (Bourhill et al. [57]), we propose that a previously unappreciated variable, that of cellular stemness in cancer (or embryonic) cells, may be a novel important factor in predicting reoviral responsiveness. It is unlikely that stemness alone will be a definitive factor in responses to reovirus, but perhaps together with a panel of other relevant variables we may eventually be able to identify which patients will benefit most from therapy that includes this or other viruses. These variables may in turn provide novel targets for therapeutic intervention for better modulation of reoviral oncolysis, as we show here in part with chemotherapeutic agents and gastric cancers. Further work will be required to evaluate more fully the utility of these proposed relationships.

## 5. Conclusions

Reovirus shows synergistic benefit when used in combination with paclitaxel or irinotecan in the treatment of gastric cancer in a murine xenograft model system. This finding may facilitate the development of effective therapeutic strategies for treating gastric cancer in patients.

## Figures and Tables

**Figure 1 viruses-15-01472-f001:**
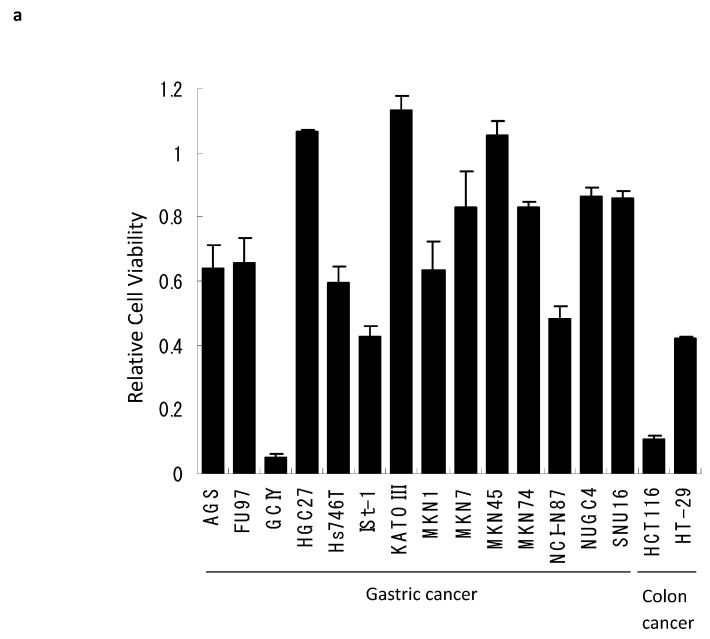
Response to reovirus is poorly correlated with levels of active Ras. (**a**) Gastric and colon cancer cell lines infected at a MOI of 10 were evaluated for viability by WST-1 assay at 72 h post infection. Bars, standard deviation (SD). (**b**) Ras activity in gastric cancer cell lines. The activated GTP-bound form of Ras was detected in most gastric cancer cell lines. Hs68 (normal human fibroblast cells) and HCT116 (colon cancer cells) were negative and positive controls, respectively. The levels of Ras activation correlated only poorly with cytolytic responses to reovirus infection.

**Figure 2 viruses-15-01472-f002:**
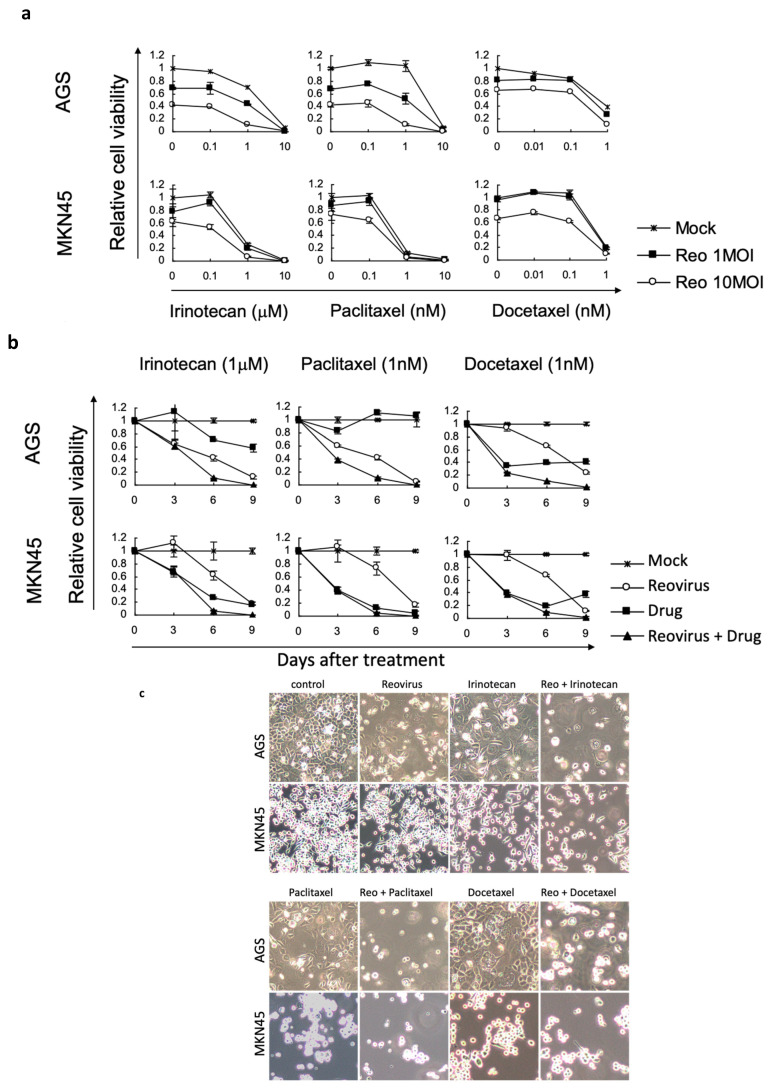
Combination effects of reovirus and chemotherapeutic agents (irinotecan, paclitaxel, and docetaxel) on human gastric cancer cell lines (MKN45, AGS, GCIY, NCI-N87). (**a**) Cells were infected with 1 or 10 MOI of reovirus and exposed to chemotherapeutic agents at the indicated concentrations. Cell viability was assessed by WST-1 assay at 6 days after treatment. Experiments were repeated at least three times and results presented as means +/− standard deviation (SD). (**b**) Time course of the combined effect of reovirus plus chemotherapeutic agents on gastric cancer cell lines. Cells were treated with 10 MOI of reovirus, chemotherapeutic agent (1 μM irinotecan, 1 nM paclitaxel, 1 nM docetaxel), or a combination of both, and cell killing efficacy was evaluated by WST-1 assay over 9 days. (**c**) Cytopathic effects of reovirus with chemotherapeutic agents. MKN45 and AGS were treated with reovirus, chemotherapeutic agents, or both (1 μM irinotecan, 1 nM paclitaxel, 1 nM docetaxel) according to the schedule described above, and photographed 5 days after treatment. ×100 magnification. (**d**) Combination indices (CI) were calculated [22] for each combination after 6 days of treatment, when differences in effect were maximal. CI values are the means of three experiments, with levels below 0.9 indicating substantial synergy, whereas a value of 1 denotes an additive effect and values above 1 indicate antagonism between the agents.

**Figure 3 viruses-15-01472-f003:**
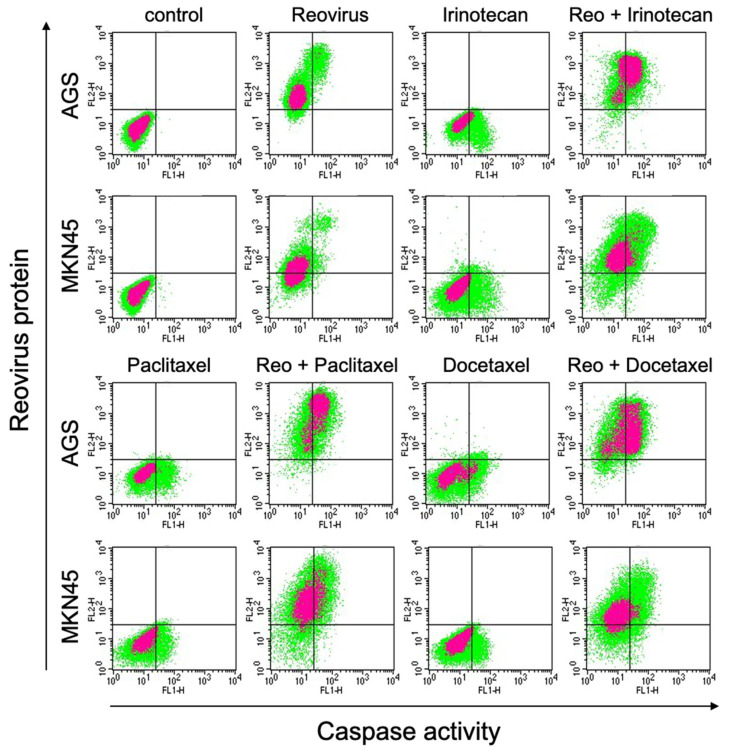
Caspase activation and reovirus protein expression. These were determined by flow cytometry in the gastric cancer cell lines MKN45 and AGS. Cells were treated with 10 MOI of reovirus, chemotherapeutic agent (1 μM irinotecan, 1 nM paclitaxel, 1 nM docetaxel), or a combination of both, and caspase activation and reovirus protein expression were evaluated by flow cytometry at 5 days post treatment. Three experiments under these conditions were performed with similar results and representative images are shown.

**Figure 4 viruses-15-01472-f004:**
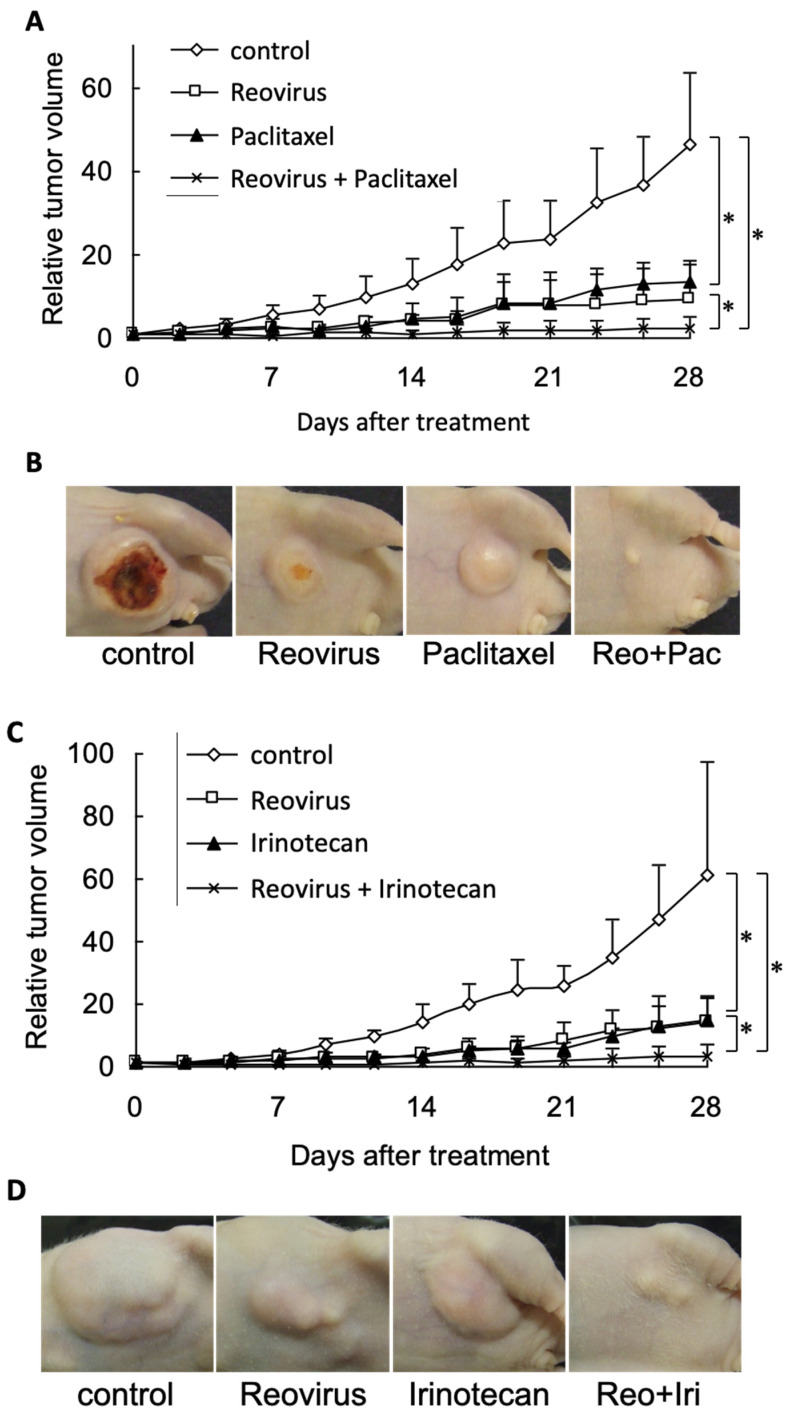
Antitumor effects of intratumorally injected reovirus and intraperitoneally administered irinotecan or paclitaxel against established flank MKN45 xenograft tumors in nu/nu mice. MKN45 cells (2 × 10^6^ cells/each) were subcutaneously injected into the left flanks of mice. Reovirus (1 × 10^8^ pfu)/body) and (**A**,**B**), irinotecan (5 mg/kg) or (**C**,**D**), paclitaxel (10 mg/kg) were administered intratumorally and intraperitoneally, respectively. Five mice were used for each group. Tumors were measured every 2 or 3 days, and tumor volume was expressed as tumor volume relative to volume at commencement of treatment. Statistical significance was defined as *p* < 0.05 (*) (Mann–Whitney *U* test). (**B**,**D**), representative tumors were photographed 28 days after treatment. (**E**) Representative histologic analysis in MKN45 xenografts treated with/without reovirus or irinotecan. Top, hematoxylin and eosin (H&E) staining. Bottom, immunohistochemistry (IHC) staining for reovirus protein. Magnification, ×200.

## Data Availability

Not applicable.

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
