# Peer review of "Modulation of Reoviral Cytolysis (I): Combination Therapeutics"

_viruses, 2023, doi:10.3390/v15071472_

Round 1
Reviewer 1 Report
Overall evaluation:
In this manuscript, the authors examined the combined effect of oncolytic reovirus with chemotherapeutic agents in gastric cancer cell lines. The use of gastric cancer cell lines is of interest and, although the idea of combination with chemotherapeutic agents is not new, such studies are clearly of interest and could lead to better use of reovirus in virotherapy.
The authors used relatively classical, yet appropriate, approaches in this study.
However, the manuscript suffers from a lack of details on different aspects (number of replicates, statistical significance, percentage of infected cells, and so on…) and complicate my interpretation of the significance of the study. I do have multiple question marks and I believe that these points should be clarified and/or discussed better.
Nevertheless, the last experiment of the manuscript (figure 4) gives a strong support to the interest of these types of study, although it was limited to tumors arising in a single cell line.
Specific comments:
•On fig. 1., the y-axis should have a label, as on subsequent figures; also, bars are SD but calculated with how many replicate? I understand that the data are presented as the mean of a certain number of replicates?
•Since there are important differences between type 3 Dearing reovirus from different laboratories, the source of the virus used should be clarified. Also, different MOI are used in the work but there is no mention on the method used to establish the virus titer to start with? Was it done in L929 cells? If this is the case, the actual MOI on the different cell lines used could be quite different. Ideally, virus titers should be determined on the same cell lines that are used in the experiments or, alternatively, the number of infected cells with the different gastric cell lines should be determined (FACS or immunofluorescence for example). If only a small percentage of the cells are infected, this could be misleading, and in fact the additive effect with the different chemotherapy agents could be even more impressive! However, in other experiments, for example 1a, the low cell killing could just reflect the low number of infected cells, and the lack of correlation with GTP-Ras becomes irrelevant? The authors should at least include more details and comment on these limitations of the study.
•Figure 2a and 2b; personally, I think that it is not necessary to indicate statistical significance, but other reviewers may think otherwise. However, the authors should at least mention the number of replicates used in SD calculation (I consider that the data are presented as the mean, although it is not indicated in the legend).
•The combination index should be better explained for readers that are less familiar with this concept, especially since the manuscript will be of interest for virologists that are probably less aware of the method and its actual calculation. Also, I believe the CI will be different depending on the time point used in the calculation and it is not mentioned at which time point it was done. Also, is it possible to calculate a statistical significance for the CI? The authors consider that everything below 1 showed synergy, this is true in theory but is a value of 0.916 really biologically relevant, or could it be considered simply additive? Is there sufficiently strong statistics to support this claim of synergy?
•Figure 3: Since this was done on day 5 (compared to day 3 in fig.1) it does indicate that not all cells are infected, especially not at the beginning of the experiment, since there is probably some viral propagation (although the authors do not examine this aspect, a kinetic of replication in the two cell lines that are examined in more detail will be quite appropriate). It will be interesting to compare the data with those obtained with a known permissive cell line such as L929 cells, is apoptosis more or less extensive than in such a classical cell model of reovirus replication? Once again, the authors do not mention how many times the experiment was done and if the differences are statistically significant, although the effect is quite clear (however, replicates sometimes generate surprising results, as we all know, and I believe that authors have done the experiment more than once, they should mention it). The authors mention “synergistic” effect but do not mention how they reach this conclusion, rather than “additive”. Also, claiming that the agents enhanced reovirus “replication” is somewhat misleading since what is actually measured is the number of infected cells exhibiting a certain level of fluorescence, does it really reflect higher production of infectious viruses?
Figure 4 is quite convincing and does support the idea of a net gain in combining reovirus with chemotherapeutic agents. However, it would have been nice to have both cell lines that were used throughout the study rather than a single cell line. Any reason for that, other than limiting the number of mice used?
Author Response
We thank the reviewer for thoughtful and constructive comments that improve the value and clarity of the manuscript significantly. Our detailed responses follow:
- The y-axis has been labelled and the M&M modified to indicate experiments were in triplicate (or greater).
- We have added a note in M&M to indicate that the virus was a gift of Dr. Lee of Dalhousie University and that titre was established using L929 cells. Regarding the issue of variable viral infectivity among different cell lines, the reviewer raises a valid point, which is that some cell lines may not support infection (low receptor levels, etc.) and others may not support viral replication leading to cell death (many possible reasons). A detailed examination of these possibilities would be a huge effort...and I have recently retired, my lab has closed and further studies by me are not possible. But we argue that the point we wish to make is a simple one, which is that some of the cell lines are exquisitely sensitive to reovirus, others are almost completely resistant and a few interesting ones show intermediate responses, with minimal correlation with levels of activated Ras protein (contrary to expectations set years ago by Dr. Lee). This leads us to the quest to enhance viral oncolysis with relevant chemotherapeutic agents, and to search for better biomarkers (the focus of the accompanying paper). We have modified the presentation of results somewhat to reflect the residual uncertainty as requested by the reviewer.
- We have clarified in M&M and in the figure legend that experiments were replicated 3 times and results are means +/- SD.
- We have added more detail regarding the combination index in the M&M and in the figure legend.
- The reviewer raises good points regarding the experiments in Figure 3. We clarified that the experiments were conducted 3 times with similar results, and we removed the word synergistic when in fact this is not strictly shown in this experiment...though the result is consistent with such an interpretation.
- The reviewer is correct - we ran out of time and funding for this particular study and presented our results to the best of our ability. We have no option for further work, but overall believe our conclusions are worthy and will be of interest to the community. We are especially keen to draw attention to the accompanying manuscript, where we identify a possible novel biomarker (cellular stemness) that may be predictive of reoviral responsiveness and that may provide opportunities for targeted delivery of virus to patients who will benefit most, and possibly even hint at strategies to modulate viral oncolytic efficiency. We argue that the two papers, though quite distinct in focus, make a nice package. The first demonstrates a weakness in utility of Ras as a biomarker for reoviral oncolysis (despite much enthusiasm over the years), but that certain chemotherapeutic drugs might enhance the effectiveness of the virus via unknown mechanisms. The second paper argues that we might be able to identify patients with tumours likely to be responsive (this idea is no longer popular and has nearly disappeared from active studies, though I think it still has merit), and that the very interesting property of stemness might be closely linked to responsiveness and perhaps even open to manipulation. I therefore hope that we will succeed in having the two papers published together, where their impact may be enhanced.
Reviewer 2 Report
This manuscript describes the effectiveness in killing gastric cancer cell lines by mammalian reovirus in combination with several chemotherapeutic agents. The authors demonstrate synergistic cytopathic effects in vitro in a panel of 14 cell lines in cell culture. Four of these were also used as subcutaneous tumor xenografts in nude mice. The authors demonstrate that the combination of reovirus and chemotherapeutic agents synergistically enhances the cytotoxicity gastric cancer cell lines in vitro and in vivo.
The studies are well described and easy to follow.
TThere is one point that the authors should describe in some more depth. How do they prevent the leakage of the virus inoculum from the injected tumor? It seems that the volume of the virus suspension 50 uL is very similar to the volume of the 5 mm tumor.
TThe figure legends to figure 2, 3 and 4 run into the main text. This is confusing. Please reformat.
Author Response
We thank the reviewer for supportive comments. For point one, we note that the injected volume of virus is approximately 1/3 of estimated tumour volume. Some leakage of injected virus from the tumour is inevitable and results in a situation similar to that encountered when viral reproduction within the tumour leads to oncolysis and release of new viral particles (which can then infect other distant tumours when these are present). We have adjusted the working in this section to make this point more clear.
For point two, we have adjusted the spacing at the end of figure legends to separate the text components, as suggested.
Reviewer 3 Report
The authors examined the ability of reovirus to induce cytolysis in gastric cancer cells and found that reovirus oncolysis and Ras activation are not closely related and that combination with chemotherapeutic agents enhances reovirus replication and apoptosis in vitro and in vivo.
Since the effects of reovirus in combination with chemotherapy have not been extensively studied, this study may provide new insights related to the combination of reovirus and chemotherapy in the treatment of gastric cancer.
There are some points to reconsider, as described below.
The references do not include very recent information. This could be improved by including more recent clinical and basic research papers.
For example,
1) Devalingam Mahalingam et al. A Phase II Study of Pelareorep (REOLYSIN®) in
Combination with Gemcitabine for Patients with Advanced Pancreatic Adenocarcinoma. Cancers 2018, 10, 160;
2) Devalingam Mahalingam et al. Pembrolizumab in combination with the oncolytic virus pelareorep and chemotherapy in patients with advanced pancreatic adenocarcinoma: a Phase 1b study. Clin Cancer Res. 2020 January 01; 26(1): 71–81.
3) Su Shao et al. Oncolytic Virotherapy in Peritoneal Metastasis Gastric Cancer: The
Challenges and Achievements. Front. Mol. Biosci. 2022, 9:835300.
4) Adil Mohamed et al. p38 Mitogen-Activated Protein Kinase Signaling Enhances Reovirus Replication by Facilitating Efficient Virus Entry, Capsid Uncoating, and Postuncoating Steps. J Virol. 2023 Feb 28;97(2):e0000923.
The advantage of reovirus among the many oncolytic viruses is its potential for use in disseminated cancer through intravenous administration. It is necessary to explain why this study is limited to intra-tumor injection and the possibility of intraperitoneal or intravenous administration.
Page 2: The authors stated that “an orphan drug designation has recently approved by the FDA for the use of reovirus in the treatment of gastric and certain other cancers”. A reference is needed to support this statement.
Page 2, references: [12-18] do not include reovirus studies. [16-20] may be [19,20].
Figure 2a and Supplementary Figure S1: The sensitivity of reovirus-resistant MKN45 to irinotecan and paclitaxel appears to be higher than that of AGS, GCIY, and NCI-N87. This finding needs to be discussed.
Figure 1 and Figure 2b: The cell viability of MKN45 infected at MOI=10 is above 1 at day 3, but drops to 0.2 at day 9. Does this mean that the infection cycle progresses very slowly in this cell line?
Figure 3: 1) The title of the figure is missing. Need to add: 2) Quantitative measurements (% of each faction) are needed to show an increasing population of caspase-activated and/or viral antigen-positive cells. 3) The authors examined the expression of viral antigens using polyclonal antibodies, but not infectious progeny virus production. Is reovirus infection in these cells productive or abortive? 4) To detect activated caspases, the authors use a labeled pan-caspase inhibitor. Does this inhibitor detect activated caspase-1? If so, this assay may detect apoptosis as well as pyroptosis (Carly DeAntoneo et al. Cells 2022, 11, 1757.). This point needs to be clarified.
Page 14 Discussion: It is necessary to specify which agents are recommended for combination therapy with reovirus.
Page 15: The authors state "(no such model currently exists with mice and gastric cancer)". Literature search shows that syngeneic models exist. The authors should review the following published studies and state if they are valid articles as syngeneic models for gastric cancer.
1) Masami Yamamoto et al. Established gastric cancer cell lines transplantable into C57BL/6 mice show fibroblast growth factor receptor 4 promotion of tumor growth. Cancer Science. 2018;109:1480–1492.
2) Guangyu Li et al. Functional characterization of a potent anti-tumor polysaccharide in a mouse model of gastric cancer. Life Sci. 2019 Feb 15;219:11-19.
3) Yan-Shen Shan et al. Establishment of an orthotopic transplantable gastric cancer animal model for studying the immunological effects of new cancer therapeutic modules. Mol Carcinog. 2011 Oct;50(10):739-50
Author Response
We thank the reviewer for thoughtful and detailed comments. We have responded as follows:
- Additional references have been included and as part of the discussion.
- We include a reference to our previous work regarding alternative methods of virus application, including directly into the digestive tract.
- A reference to FDA approval of reovirus has been included.
- The sentences in the introduction that refer to the use of chemotherapy together with oncolytic virus have been modified to indicate the earliest use with herpes virus (in addition to reovirus) and the reference numbers have been updated.
- We have acknowledged in the Results section that the responses of the cell lines to the various chemotherapeutic drugs alone varies somewhat, but is still enhanced in cytolytic efficiency by the co-administration of reovirus.
- Figure titles have been clarified. If have retired and my lab has closed, so we no longer have access to raw FACS data for recalculations. Nevertheless, we maintain that the visual presentation shows strong increases in viral protein abundance and in caspase activation, as indicated in the results section. Our previous work has shown clear evidence of productive viral infections via apoptosis, and we have included a reference to that work, while also acknowledging the possible activation of pyroptosis.
- We specify that irinotecan and paclitaxel are predicted to have superior effects in combination with reovirus in vivo.
- We were looking for a mouse model for gastric cancer that is both orthotopic AND syngeneic in an immunocompetent mouse. A renewed search has revealed a recent paper that meets these conditions, and we have added a reference in the discussion.
Round 2
Reviewer 1 Report
No further comment.